# Assessing the Costs of Home Palliative Care in Italy: Results for a Demetra Multicentre Study

**DOI:** 10.3390/healthcare10020359

**Published:** 2022-02-11

**Authors:** Gianlorenzo Scaccabarozzi, Matteo Crippa, Emanuele Amodio, Giacomo Pellegrini

**Affiliations:** 1Dipartimento Fragilità-Rete Locale Cure Palliative, ASST di Lecco, 23807 Merate, Italy; g.scaccabarozzi@asst-lecco.it; 2Fondazione Floriani, Via Privata Nino Bonnet, 2, 20154 Milano, Italy; giacomo.pellegrini@unimib.it; 3Department of Health Promotion, Maternal and Infant Care, Internal Medicine and Medical Specialties, “G. D’Alessandro”, University of Palermo, Via del Vespro 133, 90127 Palermo, Italy; emanuele.amodio@unipa.it

**Keywords:** palliative care, cost, frailty, home care

## Abstract

Background: The sustainability of palliative care services is nowadays crucial inasmuch as resources for palliative care are internationally scarce, the funding environment is competitive, and the potential population is growing. Methods: The DEMETRA study is a multicentre prospective observational study, describing the intensity of care and the related costs of palliative home care pathways. Results: 475 patients were enrolled as recipients of specialized palliative home care. The majority of recipients were cancer patients (89.4%). The mean duration of palliative care pathways was 46.6 days and mean home care intensity coefficient equal to 0.6. The average daily cost of the model with the reference variables is 96.26 euros. Factors statistically significantly associated with an increase in mean daily costs were greater dependence and extreme frailty (*p* < 0.05). Otherwise, a longer duration of treatment course was associated with a significant decrease in mean daily costs (*p* < 0.001). Conclusions: In terms of clinical and organizational management, considering the close association with the intensity and cost of the path, frailty should be systematically assessed by all facilities that potentially refer patients to home palliative care teams, and it should be carefully recorded in a standardized payment rate perspective.

## 1. Introduction

More than 63% of the people who die every year may benefit from a palliative care approach [1]. The need for palliative care is growing, mainly due to ageing of the population, and increases in cancer and non-cancer progressive diseases [2]. The delivery of palliative care should begin early in the course of life-threatening illnesses that affect patients, independently of the nature of the diseases they are affected by [3,4].

Home palliative care provided by specialized teams, for whom best characteristics have been delineated, could favor the choice of patients regarding their preferred place of care and death [5]. However, the impact of using palliative home care support in terms of cost is still poorly evaluated. Indeed, a Cochrane review highlighted that home palliative care services more than doubled the odds of dying at home, reduced symptom burdens, and significantly lowered costs [6]. Consistent with the literature [7], an extensive nationwide matched cohort study demonstrated that palliative home-based care reduced the average total medical care cost per person in the last 2 weeks of life [8]. A positive impact on facilitating home deaths, guaranteed by using end-of-life home-care programs, has been established as well [9].

Intensity of care, in terms of the frequency of consultations and access, along with the composition of the palliative care teams and the timing of referral, are measures extensively adopted for evaluating palliative home care services and their outputs [10].

Recently, it has been observed that about 60% of the Italian citizens would chose to die at home [11]. Yet, according to the Italian National Institute of Statistics, in 2018, only 37% of the deceased effectively died at their own homes. This average percentage, which varied greatly in different regions, overestimated the patients actually assisted by home palliative care units.

Italian legislation structurally introduced palliative care into the health care system through Law 38/2010. By then, many other initiatives contributed to promoting the development of palliative care services in Italy. Recently, through the Prime Ministerial Decree of 12 January 2017, with which the new essential levels of assistance were updated, home care in palliative care has become a level of assistance that must be guaranteed throughout the national territory. It was thus established that home palliative care units (HPCUs) provide both basic and specialized palliative care, thereby ensuring the unity and integration of care paths with a reference to care team on the treatment path and not on the care setting.

Other research in the Italian context [12,13] has already shown the effect of the intensity of home palliative care visits on hospital stays in the final stage of life.

Currently, limited data are available regarding the impact of the costs of these home visits and the identification of the demographic, clinical, and path-related characteristics that most influence the intensity of care and therefore these costs. Furthermore, many of the Italians territories are not adequately served by home palliative care services, and the development of those services is acutely needed [14]. Assessing the impact of palliative home-based services by monitoring process and outcome indicators is therefore paramount.

The present study investigated data from a cohort of Italian patients who received home palliative care in the last phase of their life. The primary objective was to describe the home care intensity provided by the health workers of the home palliative care teams, focusing especially on the costs of these services, and defining the factors associated with major costs.

## 2. Materials and Methods

### Study Design and Participants

The present study is based on data collected from the DEMETRA multicentre prospective observational study. The overall study enrolled 1013 patients with a chronic disease with a progressive course of any nature requiring palliative care intervention. Eligible patients were recruited from 1 May to 30 November 2017, and they were followed up for 12 months until November 2018.

The subjects were recruited at the time of admission by a specialist palliative care facility (home, hospital, and hospice) and were followed until death or upon exit from the study (drop-out or end of the monitoring period). Patients who were already on home palliative care were excluded.

This study focused only on patients with a home care service, describing the intensity of care and the related costs. The flowchart of the selection of patients evaluated in the present study is shown in Figure 1.

Details of the DEMETRA project have been reported elsewhere [15].

## 3. Data Collection

The baseline characteristics of the cohort members recorded at the time of inclusion in the study, included gender, age, cancer diagnosis (yes/no), and centre of care.

Using NECPAL, which is an internationally validated tool [16], a subject was defined in a condition of extreme fragility if they had at least two of the following conditions in the previous six months: persistent pressure ulcer (stage III−IV), recurrent infections (>1), delirium, persistent dysphagia, or falls (>2). Comorbidity was defined as the presence of at least two concomitant pathologies.

To define in detail the clinical characteristics of the patients, such as dependency in activities of daily living (ADL), data were collected with an interRAI–PC assessment instrument [17], which provides a multidimensional evaluation scale synthesizing clinical and functional, as well as social, aspects of the patient and their family. The Personal Health Profile Key (PHP) ADL scale considered in the subsequent analysis varies on a scale from 0 (no dependence) to 6 (complete dependence).

For each day of home care, information was collected regarding any home visit and the professional figures who made the visit.

The costs for each home visit were calculated, using the costs in euros per visit carried out by the specific professional figure (Table 1), overturning production costs on the accesses of the professionals, considering different wages and company costs. The economic valorisation of home visits adopted in the study considered all the costs incurred in order to produce the services, while the costs incurred for diagnostics, prosthetics, and general company costs were excluded.

The care intensity coefficient (CIA) was measured to define home care intensity. The coefficient is equal to the ratio between the number of days in which any kind of home care was provided during the period and the overall number of days the patient was assisted. 

This index is commonly used by Italian palliative care facilities to define the intensity of care, and its effectiveness has been demonstrated in recent studies [13,18]. 

### 3.1. Consent Procedure and Ethical Approval

The DEMETRA study was approved by the Ethics Committee of the Azienda Socio-Sanitaria Territoriale of Lecco (Lecco, Italy) on 1 December 2016 (Delibera 784–Reg. Pareri 223/2016), and subsequently by the institutional review boards of each participating centre. Written informed consent for participation in the study and processing of personal data were collected.

### 3.2. Statistical Analysis

Categorical variables were summarized as absolute and relative frequencies, while continuous variables were reported as mean (standard deviation (SD)) or as median (interquartile range (IQR)).

A multivariate analysis was fitted to identify the factors influencing daily costs related to visits by health professionals in home palliative care. The multicollinearity of the variables included in the model was tested by calculating the generalized variance inflation factor (GVIF). The maximum allowed value was set at 5. We applied a generalized linear model with a gamma distribution and log-link, as these are considered appropriate to model response variables with skewed distributions, such as cost variables [19,20]. Independent variables included in the model were age (<65, 65–75, 75–85, and >85 years), gender, centre (Lecco, Florence, Forlì, and Palermo), cancer diagnosis (yes/no), comorbidity (yes/no), PHP ADL (0–3, 4–6), condition of extreme fragility (yes/no), and days of home care (0–15, 16–30, 31–60, and >60 days).

All data were analysed using the R software package (version 3.6.1/2019, R Foundation for Statistical Computing, Vienna, Austria). For all of the hypotheses tested, two-tailed *p* values less than 0.05 were considered significant.

## 4. Results

A total of 475 subjects were recruited and analysed in the present study. As reported in Table 2, a large number of patients were aged 76 years or more, whereas there was an equal distribution of males and females (M/F ratio 0.94). Palliative care units of Palermo and Lecco recruited about 80% of patients involved in the study (45.9% and 33.3%, respectively). A large majority of patients were affected by cancer (89.4%) and with comorbidities (59%) and were without extreme frailty (86.9%). Patients with extreme frailty (N = 62), compared to the remaining non-frail patients, were mainly men (54.8%) and were characterized by a lower percentage of primary diagnosis of cancer (72.6% vs. 92.0%) and by a higher prevalence of comorbidities (83.9% vs. 55.2%).

In Table 3, the healthcare characteristics of the outpatient palliative care paths are reported. Overall, each patient, on average, received 9.4 (SD = 10.8) consults from palliative care physicians and 16.0 (SD = 17) visits from palliative care nurses. The total cost of visit per patients was 2681 euros (SD = 2792), which means a daily cost per patient of 75.9 euros (SD = 34.9). Patients with a primary diagnosis of cancer were on average cared for longer periods than patients with non-cancer pathologies (48.8 days vs. 30.1 days). The treatment paths for cancer patients also had a lower daily cost (73.4€ vs. 95.9€). For subjects with extreme frailty, the period of home palliative care was shorter (37.5 days vs. 48.2 days) and associated with a greater intensity of care (0.7 vs. 0.6) and higher average daily costs (91.4€ vs. 73.4€). Comparing the average costs between individuals with and without comorbidities, few differences and paths with similar durations, intensity, and costs were observed.

In Figure 2, the factors involved in determining the mean daily costs are summarized according to the multivariable analyses. The average daily cost of the model with the reference variables was 96.26 euros. The factors statistically significantly associated with an increase in mean daily costs were greater dependence (mean = 105.22 euro; 95% CI = 96.33 to 114.57) and extreme frailty (mean = 111.74 euro; 95% CI = 98.63 to 126.59). Otherwise, the longer duration of the treatment course was associated with a significant decrease in mean daily costs. Compared to short palliative home care of less than two weeks, the average daily costs were equal to 76.38, 69.09, and 66.14 euros for durations of 16–30, 30–60 days, and longer than 2 months, respectively.

## 5. Discussion

The present study estimated the costs of a home palliative care pathway using the economic value of the visits for an amount of 2681 euros (SD = 2792) per patient. This value, expressed in terms of daily cost per patient, stands at 75.9 euros (SD = 34.9).

In order to identify which characteristics play a role in defining appropriate and sustainable home care pathways, the present study analysed the intensity of care and related costs of care on a population of home palliative care recipients. Our multivariate analysis identified some of these variables, assessing their impact on the trend of care with respect to the costs.

Regardless of the unanimous indication to treat also non-cancer patients in palliative care paths [21], the majority of recipients were still cancer patients (89.4%). The prevalence of characteristics of patients affected by cancer, compared to non-cancer patients, and the median duration of home palliative care pathways (29 days), were consistent with the main non-US models of home palliative care as described in the literature [22]. On the other hand, although mean home visits increased for non-cancer patients, a cancer diagnosis seems to not have a statistically significant economic impact on the path of care. A previous Italian study [23] showed longer care paths for cancer patients, resulting from a mix of specialized and non-specialized home palliative care trajectories. The lack of systematic involvement of acute hospital wards, as well as of professionals capable of early identification and impeccable assessment of non-oncological patients with palliative care needs, may explain the higher prevalence of cancer patients and the duration of the care pathways. Indeed, previous studies have shown that systematic integration between palliative care units, the implementation of multidimensional assessment tools, and the involvement of subjects trained in early identification of people in need for palliative care may translate into a greater ability to assist non-cancer patients in palliative care [24], often with longer care paths [25], reduced hospital admissions, and emergency department visits [26]. Moreover, we have found that longer care paths tend to have a lower mean number of home visits and therefore lower mean costs. This latter finding could be caused by at least three factors: firstly, as some visits are “mandatory” for assessing the needs and interviewing the patient and family, these could impact more on short pathways than on longer ones; secondly, shorter pathways could result from late referrals. These late referred patients need to be assessed and their symptoms controlled promptly, which requires intense activities. Thirdly, the learning curve of caregiving, which plays a crucial role in assisting patients at home, requires intense activity in the beginning and final phases of home treatments: for those patients with shorter length of care pathways, nursing, empowerment and transferring of competencies are concentrated, and more qualified support is needed. We can therefore assume that the early detection of palliative care needs and, accordingly, earlier activation of home palliative care teams, could decrease the number of home visits in terms of the mean value on the care pathway. Thus, early activation of the appropriate palliative care paths could determine a higher satisfaction by the dyad patient−caregiver, with a reduced impact on cost over time in the perspective of sustainability of the national health service.

A high intensity of care is typical of Italian specialist home palliative care services, which are routinely activated for complex patients, often suffering from comorbidities (59%). These patients require a high number of visits, which can be expressed as the number of days in which home access was made compared to the total number of days of treatment.

Home care visit costs were evaluated through standardized 30 days pattern, consistent with median days of care registered among the sample. The value of 2276 (SD = 1048) euros, as the mean cost of the specialized home palliative care path, is coherent with other economic evaluations, which underlines the effects in terms of cost-savings compared to usual care [26]. 

Age has been shown to be one of the variables inversely related to the cost increase: caring for younger patients results in a higher number of home visits and therefore higher costs. Nevertheless, according to our model, the main driver of cost increase is the frailty level. Those patients with a higher degree of frailty (i.e., extreme frailty) tend to systematically receive more home visits from palliative care nurses and physicians, resulting in higher mean costs. According to our multivariable analysis, poor performances in activities of daily living are associated with an increase in mean daily costs. Indeed, frailty has recently been observed as a health feature closely related to prognosis, even in cancer patients, and a higher degree has been associated with end-of-life trajectories [24]. Among others, frailty has already been studied as a measure able to predict palliative home care intensity [27]. Similarly, an association between frailty and disability has been observed [28], with the former showing extreme susceptibility and heightened vulnerability to adverse health outcomes, and the latter showing actual disabilities in carrying out daily activities. Together, those conditions, when severe, seem to require a sharp intensification of home palliative care visits and presumably lead to an increase in burden for caregivers.

In terms of clinical and organizational management, considering the close association with the intensity and costs of the path, frailty should be systematically assessed by all facilities that potentially refer patients to home palliative care teams, and it should be carefully recorded in a standardized payment rate perspective. Measuring the level of frailty can therefore make it possible to define individualized care plans consistent with the care and assistance needs of patients, as well as to predict the duration of the end-of-life path and the related care commitment. Our results may suggest that, according to the very definition of palliative care, the level of frailty and related needs are more important than the underlying disease.

The evidence presented here may therefore contribute to the process of defining the costs of home palliative care, providing public decision makers with useful information for defining a tariff system and therefore the remuneration of the facilities that provide palliative care for account of the National Health System.

## 6. Limitations 

This study has some limitations. As we did not perform a comparison with usual care, we cannot infer that home palliative care paths are more effective in ameliorating the quality of life of the patients, nor can we derive cost-saving effects of this approach. Moreover, our sample of patients could be not fully representative of the Italian population with palliative care needs, and thus our results cannot be fully generalizable. As it is increasingly being suggested to estimate, alongside direct costs, the indirect costs of assistance paths in palliative care [29,30], our study collected data with respect to the indirect costs of home palliative care, but this information was reliable only for one unit involved. We will present that data in a future work focused on indirect costs. While aware of the difficulties in gathering information on these costs [22], in order to assess the efficiency of palliative care, we believe it is important to suggest that future investigations should identify effective solutions to overcome the obstacles that make it difficult to obtain data on indirect and societal costs.

## 7. Conclusions

This study analysed the variables associated with a higher intensity rate of care and higher costs: patient with a greater degree of frailty and those with poorer ADL performances demonstrate to receive systematically higher number of visits by professionals. Systematic frailty evaluation, performed before and during the take in charge of patients, could provide an indication about the course of the assistance and amount of resources needed to deliver palliative care at home. Longer care pathway, on the other hand, results in a lower mean intensity and cost.

As palliative home care services could provide relief and assistance on a need basis, no matter what diseases patients are affected by, and its early and appropriate integration could result in a proportionally less intense pathway, full implementation of international and national best practice, which advocate for those measures, would generate better outcomes in terms of quality of life, number of patients assisted, and lower relative costs.

## Figures and Tables

**Figure 1 healthcare-10-00359-f001:**
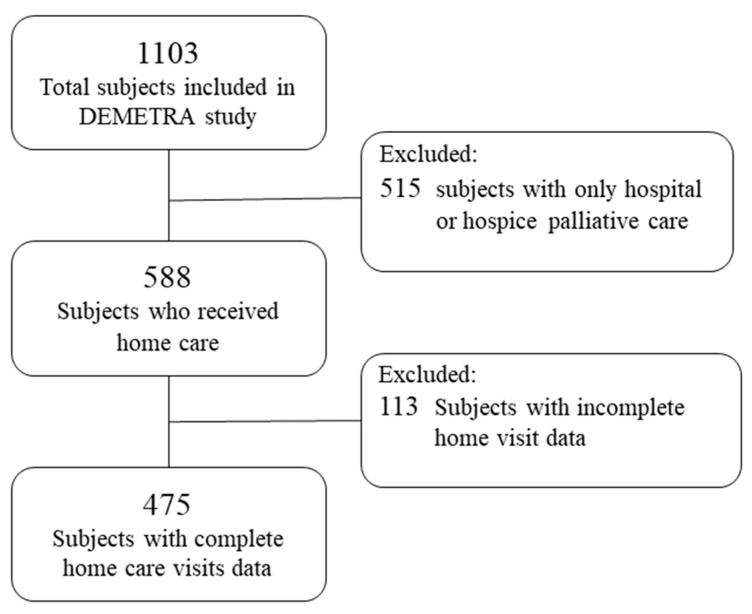
Flowchart of selection of the cohort.

**Figure 2 healthcare-10-00359-f002:**
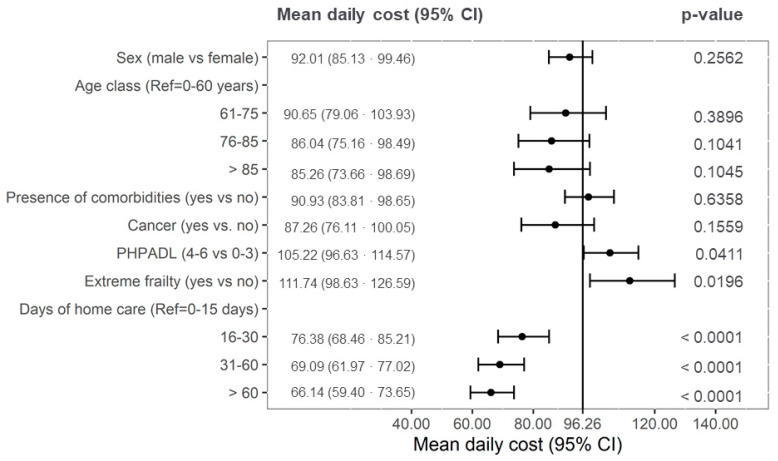
Multivariable analysis on factors involved in determining the mean daily costs in euro (the line represents the daily mean costs in the null model. All analyses have been adjusted for study centre).

**Table 1 healthcare-10-00359-t001:** Costs of home visits for health workers.

Medical Professional	€/Home Visit
Palliative physician	120.00
Dietician	60.00
Social worker	60.00
Psychologist	70.00
Physiatrist	120.00
Nurse	60.00
Physiotherapist	60.00
Speech therapist	60.00
Non-palliative physicians	120.00
Social health worker	40.00

**Table 2 healthcare-10-00359-t002:** Baseline characteristics of patients involved in the study, overall and stratified for the presence of factors of extreme fragility.

	Total Study Cohort (N = 475)	Patients in Extreme Frailty Condition (N = 62)	Patients in Not Extreme Frailty Condition (N = 411)
	**N (% of total)**	**N (% of total with frailty)**	**N (% of total not frailty)**
**Age, median (IQR)**	78.0 (12.2)	81.0 (11.0)	78.0 (16.0)
**Age (class)**			
0–60	50 (10.5)	5 (8.1)	45 (10.9)
61–75	146 (30.7)	12 (19.4)	133 (32.6)
76–85	163 (34.3)	22 (35.5)	141 (34.3)
≥85	116 (24.4)	23 (37.1)	92 (22.4)
**Sex**			
M	230 (48.4)	34 (54.8)	196 (47.7)
F	245 (51.6)	28 (45.2)	215 (52.3)
**Centre**			
Florence	52 (10.9)	1 (1.6)	51 (12.4)
Forlì	47 (9.9)	14 (22.6)	33 (8.0)
Lecco	158 (33.3)	6 (9.7)	152 (37.0)
Palermo	218 (45.9)	41 (66.1)	175 (42.6)
**Primary cancer diagnosis ***			
Yes	423 (89.4)	45 (72.6)	378 (92.0)
No	50 (10.6)	17 (27.4)	33 (8.0)
**Comorbidities ***			
Yes	279 (59.0)	52 (83.9)	227 (55.2)
No	194 (41.0)	10 (16.1)	184 (44.8)
**Extremely Frailty condition ***			
Yes	62 (13.1)		
No	411 (86.9)		


* 2 missing.

**Table 3 healthcare-10-00359-t003:** Healthcare characteristics of the outpatient palliative care paths, overall and stratified by cancer, comorbidities and extreme frailty.

	Total Study Cohort	Cancer		Comorbidities		Extreme Frailty	
	N = 475	Yes (N = 423)	No (N = 50)	Yes (N = 279)	No (N = 194)	Yes (N = 411)	No (N = 62)
	Mean (SD)						
**Days of home care per patient**	46.6 (54.9)	48.8 (56.3)	30.1 (39.1)	45.7 (52.0)	48.4 (59.1)	37.5 (48.8)	48.2 (55.8)
**Number of Palliative care physician visits per patient**	9.4 (10.8)	9.9 (11.1)	5.8 (6.7)	8.9 (9.5)	10.3 (12.5)	7.5 (8.7)	9.8 (11.1)
**Number of Nurse visit per patient**	16.0 (17.0)	16.3 (17.1)	14.1 (15.7)	15.6 (15.9)	16.8 (18.4)	14.7 (17.2)	16.3 (17.0)
**Days of care with at least one visit**	25.7 (28.6)	26.5 (29.3)	19.5 (21.9)	25.0 (26.0)	26.9 (32.2)	22.4 (24.9)	26.3 (29.2)
**Home care intensity coefficient**	0.6 (0.2)	0.6 (0.2)	0.7 (0.2)	0.6 (0.2)	0.6 (0.2)	0.7 (0.2)	0.6 (0.2)
**Total cost of visits per patient, €**	2681 (2792)	2762 (2856)	2097 (2117)	2648 (2624)	2755 (3026)	2387 (2480)	2738 (2837)
**Daily cost of visits per patient, €**	75.9 (34.9)	73.4 (33.0)	95.9 (40.9)	77.2 (35.2)	73.7 (33.7)	91.4 (43.2)	73.4 (32.5)
**Cost of visits per patient for 30 days of care, €**	2276 (1048)	2202 (991)	2876 (1228)	2316 (1056)	2211 (1011)	2742 (1296)	2202 (975)

## Data Availability

The data presented in this study are available on request from the corresponding author.

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
