# Peer review of "Assessing the Costs of Home Palliative Care in Italy: Results for a Demetra Multicentre Study"

_healthcare, 2022, doi:10.3390/healthcare10020359_

Round 1

Reviewer 1 Report

The present paper is about an important topic of home palliative care that follows a cohort of Italian patients

A better discussion of the multivariate analysis could be done. For example the collinearity of correlation between comorbidity and cancer and other variables 

I would suggest to reference other studies of home palliative care at the international level, this could situate better the study in relation to the international literature, to improve the understanding of the methodological section as well as the discussion of the results.

The study could be improved with a more detailed discussion of the factors extreme frailty and dependence (PHPADL) 

I have also some questions of the cofounding factors and the multicollinearity of variables such as cancer and co-morbidity. 

For example, the use of the NECPAL tool could be a point to compare with other studies. 

See this recent reference

Turrillas P, Peñafiel J, Tebé C, et alNECPAL prognostic tool: a palliative medicine retrospective cohort studyBMJ Supportive & Palliative Care Published Online First: 16 February 2021. doi: 10.1136/bmjspcare-2020-002567

Or also:

Gómez-Batiste, X., Martínez-Muñoz, M., Blay, C., Amblàs, J., Vila, L., Costa, X., Espaulella, J., Villanueva, A., Oller, R., Martori, J. C., & Constante, C. (2017). Utility of the NECPAL CCOMS-ICO© tool and the Surprise Question as screening tools for early palliative care and to predict mortality in patients with advanced chronic conditions: A cohort study. Palliative Medicine, 31(8), 754–763. https://doi.org/10.1177/0269216316676647

  Walsh, R. I., Mitchell, G., Francis, L., & van Driel, M. L. (2015). What Diagnostic Tools Exist for the Early Identification of Palliative Care Patients in General Practice? A systematic Review. Journal of Palliative Care, 31(2), 118–123. https://doi.org/10.1177/082585971503100208

One important point is to improve the discussion of the results, the multivariate analysis, and to present more details of how it was done

There is in the bibliography a systematic use of Italian for dates, month and other aspects (citato, settembre, etc. ) This is a detail, but given the publication is in English, this should be rectified.

Author Response

REVIEWER 1

The present paper is about an important topic of home palliative care that follows a cohort of Italian patients

A better discussion of the multivariate analysis could be done. For example the collinearity of correlation between comorbidity and cancer and other variables 

Thank you very much for your considerations. The multicollinearity of the variables included in the model was tested by calculating the generalized variance inflation factor (GVIF).

The maximum allowed value has been set at 5. No multicollinearity was recorded for any of the variables of the multivariate model. We have updated the manuscript with this information (pag.4 - Statistical Analysis)

[John Fox & Georges Monette (1992) Generalized Collinearity Diagnostics, Journal of the American Statistical Association, 87:417, 178-183, DOI: 10.1080/01621459.1992.10475190 ]

I would suggest to reference other studies of home palliative care at the international level, this could situate better the study in relation to the international literature, to improve the understanding of the methodological section as well as the discussion of the results.

Thanks for your suggestions. We updated references and discussion sections.

The study could be improved with a more detailed discussion of the factors extreme frailty and dependence (PHPADL) 

I have also some questions of the cofounding factors and the multicollinearity of variables such as cancer and co-morbidity.

One important point is to improve the discussion of the results, the multivariate analysis, and to present more details of how it was done.

Your comments have been especially useful to improve analysis and to improve discussion section, suggesting three explanations about the different distribution of the costs. We have also expanded the discussion on frailty and disability, recalling a new bibliographic reference and indicating potential consequences in terms of increased burden for caregiver.

There is in the bibliography a systematic use of Italian for dates, month and other aspects (citato, settembre, etc. ) This is a detail, but given the publication is in English, this should be rectified.

We apologize for these mistakes. We have changed the references and checked for further typos.

Reviewer 2 Report

I read with interest the paper by Scaccabarozzi et al. However, I have some concerns.

It is unclear how authros calculated the cost of home visits. These costs are overbudget in the real world, where the Heath care System reimbourses 60 euro/die for speciliazed palliative care or 30 euro /die for 1 level assistance. Also difffernces between palliative physician and physician are unclear. In Results only costs regarding palliative care care physicians and nurses are reported, accounting for a total of 1740 euro

Data were all introduced in a multivariate analysis, without screening the variables first an unisariate analysis.

  Discussioni is disordered and is not fluent. Please, start the discussion with the principal findings of the study, without reporitng items already presented in introduction to explain the reason  of the study: “The present study showed that…..

The discussion should be based on the outomes of the study, rather than being a general view.

Conclusion is repetitive and long enough. It should be 5-10 lines.

Minor comments (regarding English..)

The course of sentences in Introduction is poorly fluent and there are many repetitions.

30 “non-communicable diseases” should be non cancer progressive diseases

33 Change the sentence Despite these facts etc.

“Home palliative care provided by specialized teams, whom best 33 characteristics have been delineated, could favour the choice of patients regarding their 34 preferred place of care and death (5). However, the impact of using palliative home care support in 35 terms of costs is still poorly evaluated”. Indeed, a Cochrane review highlighted that 36 home palliative care services more than doubled the odds of dying at home, reduce symp- 37 tom burdens and significantly lower the costs (6)

63 On hospital stays (delete in)

70 In the final stage of life (at the last phase of their life)

77 having a chronic disease

83 who were already on home palliative care

94 Using NECPAL, that is a validated tool….

114 is equal to theratio between  …………. And

115 was assisted

143 Palliative care units

180 The data of patients ………   is consistent with the main non-US models

183 On the other hand

189 A previous

194 home palliative care units (to do not confond with palliative care unit in hospital),

196 may result

202 the number of home visits

227 The study has some limitations

Author Response

REVIEWER 2

I read with interest the paper by Scaccabarozzi et al. However, I have some concerns.

It is unclear how authros calculated the cost of home visits. These costs are overbudget in the real world, where the Heath care System reimbourses 60 euro/die for speciliazed palliative care or 30 euro /die for 1 level assistance. Also difffernces between palliative physician and physician are unclear. In Results only costs regarding palliative care care physicians and nurses are reported, accounting for a total of 1740 euro

Thank you for your comments and suggestions. Table 1. Costs of home visits per health workers provide information we used to estimate home accesses, considering the standard palliative care teams which require Palliative Physician (compulsory in Specialized palliative care level of assistance) and sometimes can integrate physicians with other specialties or non-specialized physicians. To clarify, we improved the labels.

Data were all introduced in a multivariate analysis, without screening the variables first an unisariate analysis.

Discussioni is disordered and is not fluent. Please, start the discussion with the principal findings of the study, without reporitng items already presented in introduction to explain the reason of the study: “The present study showed that…..

We sincerely thank you for your suggestion. We have edited the discussion according to your evaluable comments.

The discussion should be based on the outomes of the study, rather than being a general view.

Conclusion is repetitive and long enough. It should be 5-10 lines.

Thanks for the suggestion, which we have accepted by reviewing discussion and conclusions.

Minor comments (regarding English..)

The course of sentences in Introduction is poorly fluent and there are many repetitions.

30 “non-communicable diseases” should be non cancer progressive diseases

33 Change the sentence Despite these facts etc.

“Home palliative care provided by specialized teams, whom best 33 characteristics have been delineated, could favour the choice of patients regarding their 34 preferred place of care and death (5). However, the impact of using palliative home care support in 35 terms of costs is still poorly evaluated”. Indeed, a Cochrane review highlighted that 36 home palliative care services more than doubled the odds of dying at home, reduce symp- 37 tom burdens and significantly lower the costs (6)

63 On hospital stays (delete in)

70 In the final stage of life (at the last phase of their life)

77 having a chronic disease

83 who were already on home palliative care

94 Using NECPAL, that is a validated tool….

114 is equal to theratio between  …………. And

115 was assisted

143 Palliative care units

180 The data of patients ………   is consistent with the main non-US models

183 On the other hand

189 A previous

194 home palliative care units (to do not confond with palliative care unit in hospital),

196 may result

202 the number of home visits

227 The study has some limitations

Thanks for the opportunity to correct errors and ameliorate fluency of the sentences.

Reviewer 3 Report

This manuscript is well written and analyzes the daily costs of home palliative care in Italy. The authors calculate an average daily cost of around 70 euros. In a multivariate analysis extreme frailty and greater dependence as assessed by ADL score were associated with increased costs, a longer duration of the treatment with lower costs. A strength of this analysis is the multivariate analysis of the effort of home palliative care. Despite a clear research hypothesis several major issues need to be improved.

First, the statistical analysis:
The authors should further delineate the calculation of the costs in table 1. Are these amounts professional fees accounted to the social insurance?

The authors should clarify why they used once the median and once the mean? Was the age normally distributed? Why is the age in Table 2 presented as mean with IQR?

Moreover, it could be interesting if there are differences between patients being extremely frail or not and patients with and without cancer. Why not compare these two groups in table 2?

Table 3 presents the main parameters comparing medians with means, which is not really important information. I suggest to improve the message of table 3 by a comparison of the main parameters using several groups of patients like cancer vs. non-cancer, cardiac vs. neurologic. Or Lecco vs. Palermo vs. Florence vs. Forlì.

In Table 3 are reports only visits by palliative care physician or nurses? Were the patients visited by other professionalists, specially cancer patients may have been visited by dieticians, like physiatrists, social workers, …?

Again, in table 3 the used descriptive statistics are varying. The mean number of visits vs. the median costs.

The authors could provide more information about the distribution of the palliative visits. Are they conducted on a regular basis or as needed? Are there more visits at the beginning of the treatment (which could explain lower costs with longer treatments)?

Second, the discussion needs to be improved:

In the discussion the different strength of the results should be weighted carefully. The PHPADL reaches significance only just as compared to the very strong findings of a longer treatment. Moreover, the authors should discuss possible reasons of the fact that the longer a treatment lasts the lower the costs get. Are these patients less ill than those dying earlier? Are they cared for better and need less frequent visits? Or are they cared for worse? Any hints from literature?
Again, it would be interesting if different underlying diseases show different palliative paths and costs.

Minor comments: Half of the references are older than 5 years. The authors use an adequate number of self-citations.
Page 1, line 33f: This sentence is very long and hard to understand. The authors could add a new sentence after citation 5.

Footer of table 2 and 3: please mention abbreviations like IQR, SD and the statistical methods
Page 7, line 188: “… a high intensity rate of care (0.6)” what does 0.6 stand for?
Page 7, line 189: A previous study?
Page 8, line 251: this sentence needs to be improved.

Given the stated above the manuscript needs to be improved substantially, but could contributed relevant data to the field.

Author Response

REVIEWER 3

A strength of this analysis is the multivariate analysis of the effort of home palliative care. Despite a clear research hypothesis several major issues need to be improved.

First, the statistical analysis: The authors should further delineate the calculation of the costs in table 1. Are these amounts professional fees accounted to the social insurance?

Thanks for the opportunity to better clarify the methods we applied to valorise costs. Valorisation process we applied did not consider the costs accounted to the social insurance, since Italian health care system in Italy is mainly built on a universalistic model. Rather, we have estimated the costs for each visit, as indicated in the table, overturning the production costs resulting from the analytical accounting on the accesses of the professionals, considering different wage and public company cost.

The authors should clarify why they used once the median and once the mean? Was the age normally distributed? Why is the age in Table 2 presented as mean with IQR? Moreover, it could be interesting if there are differences between patients being extremely frail or not and patients with and without cancer. Why not compare these two groups in table 2?

Thanks for this correct note. Age is not normally distributed. We therefore corrected table 2 by replacing the value of the mean(SD) with median (IQR).

As correctly suggested we have also included in table 2 a comparison between groups based on the presence of extreme fragility. We have therefore updated the entire table in the manuscript.

Table 3 presents the main parameters comparing medians with means, which is not really important information. I suggest to improve the message of table 3 by a comparison of the main parameters using several groups of patients like cancer vs. non-cancer, cardiac vs. neurologic. Or Lecco vs. Palermo vs. Florence vs. ForliÌ€. In Table 3 are reports only visits by palliative care physician or nurses? Were the patients visited by other professionalists, specially cancer patients may have been visited by dieticians, like physiatrists, social workers, …? Again, in table 3 the used descriptive statistics are varying. The mean number of visits vs. the median costs.

Thanks for your interesting considerations. We agree it is possible to improve table 3. Following your suggestion we have therefore selected only the mean values (with sd) for the whole cohort and stratifying by presence/absence of cancer, comorbidity and extreme fragility. We have therefore updated the entire table in the manuscript.

The authors could provide more information about the distribution of the palliative visits. Are they conducted on a regular basis or as needed? Are there more visits at the beginning of the treatment (which could explain lower costs with longer treatments)?

Thank you for your consideration which allow us to provide better explanation regarding palliative care service delivering. Visits are conducted both on regular basis as well as patients or family need it: adaptability and flexibility are two main cornerstones of palliative care. In general terms, first assessment visit is mandatory and regular visits are usually planned. Nonetheless, according to the progress of the needs and the performances on different indicators monitored by the team reduction or intensification of the home care visits even just for new assessments are common in practice and in our sample as well.

Regarding length and intensity of treatments resulting in different costs distributions, which is an interesting observation, we are not able to provide a definitive response. However, we may presume that different causes are at stake: firstly, coherently with the point A, since some visits are “mandatory”, those could impact more on short pathways than on longer ones; secondly, shorter pathways could result from late referrals – which is a common problem many healthcare systems are tackling, even where PC are well established and integrated. These late referred patients need to be assessed and their symptoms controlled promptly, which require intense activities. Thirdly, learning curve of caregiving, which plays crucial role in assisting patients at home, require intense activity in the beginning phases of the home treatment: for those patients with shorter length of care pathways, nursing and transferring of competencies are concentrated and more qualified support is needed. All these drivers could impact on intensity of care and, therefore, on costs.

All the previous consideration have been reported in the Discussion section of the revised manuscript.

Second, the discussion needs to be improved:

In the discussion the different strength of the results should be weighted carefully. The PHPADL reaches significance only just as compared to the very strong findings of a longer treatment. Moreover, the authors should discuss possible reasons of the fact that the longer a treatment lasts the lower the costs get. Are these patients less ill than those dying earlier? Are they cared for better and need less frequent visits? Or are they cared for worse? Any hints from literature?
Again, it would be interesting if different underlying diseases show different palliative paths and costs.

Thanks to your valuable suggestions. According to your considerations and those made by the other Reviewers we have rewritten the discussion that, at our opinion, is significantly improved.

Minor comments: Half of the references are older than 5 years. The authors use an adequate number of self-citations. Page 1, line 33f: This sentence is very long and hard to understand. The authors could add a new sentence after citation 5.

Footer of table 2 and 3: please mention abbreviations like IQR, SD and the statistical methods. Page 7, line 188: “… a high intensity rate of care (0.6)” what does 0.6 stand for?

Page 7, line 189: A previous study?

Page 8, line 251: this sentence needs to be improved.

Thanks for your comments. We have modified references and ameliorated the sentences you highlighted.

Round 2

Reviewer 2 Report

no changes required

Author Response

We appreciated your qualified help. Thank you.

Reviewer 3 Report

Thank you for adressing the raised major issues. I have only some minor points left:

@ Table 2: why not "% extremly frail patients" and "% mildly frail patients"?

line 197: did you mean "on the other hand"?

@ Table 3: a comparison between the two groups with median (IQR) and p-value would be have been fine. However, I feel the article being suitable for publication even without this further analysis.

Author Response

Thank you for adressing the raised major issues. I have only some minor points left:

@ Table 2: why not "% extremly frail patients" and "% mildly frail patients"?

Thank you for the notes on the language, which we have improved according to your suggestion.

line 197: did you mean "on the other hand"?

Thank you for the comment on table 2, certainly pertinent to the enrolled cases. However, the questionnaire used in the DEMETRA project does not allow us to define the degrees of frailty, but rather identifies the conditions of "extreme frailty" with respect to the rest of the sample. We therefore consider it more correct methodologically to maintain the distinction "extreme frailty" versus "not extreme frailty".

@ Table 3: a comparison between the two groups with median (IQR) and p-value would be have been fine. However, I feel the article being suitable for publication even without this further analysis.

Thanks for the suggestion about table 3. We agree with you about the possibility of developing a further comparison. However, since it is already a fairly complex table, we believe it is more usable in its current form.